# Challenges for Natural Hydrogels in Tissue Engineering

**DOI:** 10.3390/gels5020030

**Published:** 2019-05-29

**Authors:** Esmaiel Jabbari

**Affiliations:** Biomimetic Materials and Tissue Engineering Laboratory, Department of Chemical Engineering, University of South Carolina, Columbia, SC 29208, USA; jabbari@cec.sc.edu; Tel.: +1-803-777-8022; Fax: +1-803-777-0973

**Keywords:** protein-based hydrogel, hierarchical structure, cell encapsulation, cell function, tissue engineering

## Abstract

Protein-based biopolymers derived from natural tissues possess a hierarchical structure in their native state. Strongly solvating, reducing and stabilizing agents, as well as heat, pressure, and enzymes are used to isolate protein-based biopolymers from their natural tissue, solubilize them in aqueous solution and convert them into injectable or preformed hydrogels for applications in tissue engineering and regenerative medicine. This review aims to highlight the need to investigate the nano-/micro-structure of hydrogels derived from the extracellular matrix proteins of natural tissues. Future work should focus on identifying the nature of secondary, tertiary, and higher order structure formation in protein-based hydrogels derived from natural tissues, quantifying their composition, and characterizing their binding pockets with cell surface receptors. These advances promise to lead to wide-spread use of protein-based hydrogels derived from natural tissues as injectable or preformed matrices for cell delivery in tissue engineering and regenerative medicine.

## 1. Introduction

The application of hydrogels is a very exciting prospect in tissue engineering as a matrix for delivery and retention of cells and growth factors within the site of injury [1,2,3,4]. Oxygen, nutrients, proteins and other biomolecules readily permeate through the water-swollen network of hydrogels to nourish the encapsulated cells [5,6]. Synthetic, as well as natural hydrogels, are used extensively as preformed or injectable matrices in cellular tissue constructs [7,8,9,10,11,12,13]. Although synthetic hydrogels provide matrices with tunable properties, they lack the bioactive amino acid sequences required for adhesion, growth, and maturation of the encapsulated cells [14,15]. As a result, hydrogels derived from natural sources like animal or human cadaver tissues, hereafter referred to as biogels, are used extensively in tissue augmentation, repair, and regeneration. There has been an explosion of new biopolymers in the past few decades derived from proteins, lipids, nucleic acids, and carbohydrates [16]. Among these, protein-based biogels are used extensively in medicine because they resemble the natural extracellular matrix (ECM) of mammalian tissues. Protein-based materials possess a hierarchical structure in their native state which profoundly affects their function. As protein biopolymers are isolated from their tissue of origin by physical, mechanical, chemical, or biochemical methods, some of these microstructures are lost during processing, thus giving rise to new materials with relatively unpredictable biological and cellular properties. This short review highlights the need to better understand the microstructure of hydrogels derived from natural tissues and the effect of these microstructures on the fate of the encapsulated cells. 

## 2. Sources of Protein-Based Biopolymers and Their Isolation Methods

Biopolymers from natural sources include plasma (fibrinogen) [17], tissues (collagen and gelatin) [18,19], skin appendages (wool, hair and feather) [20], cocoons of the larvae (sericin and fibroin) [21], and ECM-based biopolymers produced by cultured cells in vitro [22]. Fibrinogen is isolated from blood plasma by affinity extraction [23]. Collagen is extracted from mammalian tissues using solubilizing agents in the absence of or low enzymatic digestion such that the triple-helical and fibrillar structures of collagen are conserved [24]. Conversely, skin tissue is treated with strong acids or bases at high temperature and pressure to partially break down the triple-helical and fibrillar structures of collagen to produce water-soluble, high-molecular-weight gelatin [25]. In another approach, tissues are chopped into small pieces, decellularized and digested with collagenase and/or pepsin to produce a water-soluble, low-molecular-weight product biopolymer [26]. In the case of sericin and fibroin, strong hydrogen bonding solvents are used to solubilize the proteins in an aqueous solution [27,28]. For wool, hair and feather, a combination of reducing agent, solubilization and stabilizing agents are used to dissolve the protein in an aqueous solution [11]. As most proteinaceous materials are amphiphilic, they need to be solubilized and stabilized in aqueous solution. 

## 3. Structure of Protein-Based Hydrogels

A common characteristic of protein-based biopolymers and biogels is their ability to form structures at multiple length scales from nano- to micro-, meso- to macroscales. Their hierarchical structure is rooted in the unique sequence of amino acids of protein-based biopolymers and the formation of α-helical, β-sheet, and random coil secondary structures at the nano-scale as well as other structures at micro-/macro-scales. In the following paragraphs, we compare isolation protocols and structural features of collagen and gelatin derived from skin, digested ECM from articular cartilage tissue, and β-keratin from feather.

Collagen is extracted from skin tissue by first removing fat deposits followed by stripping non-collagenous proteins and polysaccharides with sodium acetate, followed by extraction of collagen from the tissue with sodium citrate [29]. Sodium citrate as a mild base serves as a solubilizing agent for collagen molecules, thus the triple-helical and fibrillar structure of collagen remains relatively intact after isolation. Conversely, gelatin is produced by breaking down the collagenous proteins in the skin tissue using strong acids or based at high temperatures and pressures which affects the hierarchical structure of collagen molecules [25]. Collagen has been used extensively in clinical applications as a biogel for cell and growth factor delivery [30,31,32]. Both collagen and gelatin form honeycomb-like microstructures when cast into three-dimensional scaffolds, with collagen more fibrous at the nanoscale compared to gelatin [10,33]. If we had more information and understanding about the exposed/available peptides in gelatin scaffolds for interaction with receptors on the surface of encapsulated cell to assess in vivo cellular performance, gelatin could be used as a substitute for collagen at a significantly reduced cost in many clinical applications. 

Articular cartilage is structurally and functionally composed of multiple distinct zones including the superficial, middle and calcified zones, with each zone exhibiting a defined ECM composition and organization [34,35]. The superficial zone which lies at the joint surface is characterized by thin collagen fibrils oriented parallel to the articular surface. The middle zone is composed of thick collagen fibrils with random orientation whereas in the calcified zone collagen fibers are oriented perpendicular to the articular surface. The calcified zone is at the interface between the uncalcified hyaline cartilage and subchondral bone with unique mineralized matrix composition. In articular cartilage, the compressive modulus increases from the articular surface to the deep zone [36].

A major challenge in articular cartilage regeneration is creating hydrogels that mimic the physical and biochemical properties of different cartilage zones. In that regard, digested and decellularized articular cartilage is used as an injectable matrix in cartilage regeneration [13,35]. The articular cartilage is cut into small pieces, frozen in liquid nitrogen, milled, and sieved to obtain <1 mm fragments. Next, the fragments are dispersed in Tris buffer supplemented with Triton X-100 and sonicated to decellularize the cartilage. Then, the fragments are maintained in a nuclease solution to degrade the DNA and RNA, and lyophilized. Next, the fragments are dispersed in a 0.01 M hydrochloric acid solution supplemented with pepsin to digest and solubilize the cartilage fragments [26]. After deactivation of pepsin, methacrylic anhydride is added to the solution under stirring to allylate the digested cartilage. After allylation, the mixture is dialyzed against deionized water to remove unreacted agents and lyophilized to generate a foam. The functionalized, digested articular cartilage is dispersed in a buffered solution; the suspension is dispensed into a mold and crosslinked by ultraviolet (UV) irradiation to generate an articular cartilage-based in situ-gelling hydrogel. Scanning electron microscopy (SEM) images in Figure 1 demonstrate that the decellularized bovine articular cartilage (left image) and the digested decellularized allylated cartilage (right image) have similar honeycomb-like structure. Although the composition and microstructure of decellularized and UV-gelled, allylated, digested, decellularized articular cartilage are identical (except for allylation), more information about the differences in available/exposed peptides or binding pockets for interaction with receptors on the surface of encapsulated cells is needed for using the in situ-gelling, digested cartilage tissue in clinical applications. 

The study of cell behavior on protein-based hydrogels is of great interest to regenerative medicine. In that regard, keratin is isolated from a bird feather by a combination reduction, solubilization and stabilization [11]. More than 98% of the feather extract is β-keratin with a relatively short chain length (~100 amino acids for chicken feather) compared to other ECM-based biopolymers [11]. A feather has an intriguing hierarchical structure as shown in Figure 2 [37,38]. The β-keratin chains in feathers consist of a β-sheet flanked by two random coils at N- and C-terminals. The first step in the process of assembly is β-sheet dimer formation between two β-keratin chains (Figure 2a,b). Next, the dimers assemble by stacking to form a β-keratin fibril (Figure 2c). Then, the fibrils assemble by inter-molecular interaction between the N- and C-terminal chains on either side of the β-sheet in β-keratin molecules to form a β-keratin 2D fiber sheet (Figure 2d,e). Finally, the β-keratin fiber sheets assemble to form a β-keratin 3D fibrous matrix. Extensive secondary interactions, including hydrogen bonding, electrostatic and hydrophobic interactions are involved in the formation of β-sheet regions in individual β-keratin molecules, dimer formation between two β-keratins, and fibril formation [37]. The random coil regions in the 2D fiber sheets (Figure 2d) and 3D fibrous matrix (Figure 2f) are held together by a network of disulfide (S–S) bridges as well as secondary interactions between the chains. The existence of sheet-like structures is indicative of long-range interactions at the macro-scale spanning the entire dimensions of the specimen. Keratin has a relatively high fraction of cysteine residues compared to other proteins varying from 7% in a feather to 15% in wool keratin [20]. The disulfide crosslinks in combination with hydrogen bonding impart high strength to keratin-based tissues [39].

The keratin was extracted from feather barbs using a mixture of tris(2-carboxyethyl) phosphine (TCEP) as a reducing agent to convert S–S disulfide bonds to sulfhydryl groups (–SH), urea as a protein solubilizing agent to disrupt intra- and inter-molecular hydrogen bonds, and sodium dodecyl sulfate (SDS) as a surfactant for stabilization and transport of keratin molecules within micelles to the aqueous solution [40]. For keratin isolation, the feather is cleaned by soaking in ether followed by washing with soap water, dried and cut into small pieces [11]. The cut pieces are immersed in deionized water (DI) containing 0.5 M sodium dodecyl sulfate (SDS), urea, and tris(2-carboxyethyl) phosphine (TCEP) and the pH is adjusted to 6.5. Next, the suspension is heated to 50 °C and maintained at that temperature for 6 h under constant stirring to dissociate the disulfide bonds. 

The solution is filtered and centrifuged to remove undissolved fragments, dialyzed against DI water to remove any remaining SDS, TCEP, and urea. Then, the solubilized keratin in sodium phosphate buffer is supplemented with 1 mM TCEP and reacted with O-(2,4,6-trimethylbenzenesulfonyl)-hydroxylamine (MSH) for 20 min under stirring to convert sulfhydryl group of cysteine (–SH group) to dehydroalanine (Dha) [41]. Then, allyl mercaptanol is added to the reaction mixture to allylate Dha to S-allyl cysteine and form keratin allyl thioether. The allyl-functionalized keratin is dialyzed against deionized water and lyophilized to form a powder. The functionalized feather keratin is dispersed in a buffered solution, the suspension is dispensed into a mold and crosslinked by ultraviolet (UV) irradiation to generate a keratin-based hydrogel [11].

A variety of techniques including circular dichroism (CD), infrared spectroscopy, gel electrophoresis, thermogravimetric analysis, molecular weight fractionation by dialysis, rheometry, electron microscopy (SEM), degradation, and cell adhesion experiments are used to characterize physical and biochemical properties of the feather keratin hydrogel [11]. Figure 3 shows CD results, SEM images, and cell adhesion properties of the keratin hydrogel. CD absorption bands with peak positions at 208 and 220 nm in Figure 3a, attributed to α-helical structures, are observed in the spectrum of the feather hydrogel (green and light blue curves) [42]. The α-helical absorption bands emanate from the periodicity of β-keratin fibrils formed by stacking of β-keratin dimers (see Figure 2c). A broad band with peak position between 200 and 235 nm, attributed to β-sheet structures, is also observed in the spectrum of the hydrogel [43]. The fraction of α-helical and β-sheet structures in the hydrogel, based on analysis of the spectral data, is 38 ± 6% and 9 ± 8%, respectively. 

Functionalization of feather keratin with allyl groups does not significantly change the fraction of α-helical and β-sheet structures (compare purple and light blue curves in Figure 3a) and the CD intensity only depended on concentration of feather keratin (compare red and purple or green and light blue in Figure 3a). Based on CD results, the possible microstructures for solubilized feather keratin are single keratin chain β-sheets, β-sheet dimers, fragments of β-keratin fibrils, and fragments of β-keratin 2D sheets, as shown in Figure 4. The absence of 2D fiber sheets (Figure 2d) and 3D fibrous matrix (Figure 2f) is indicative of the lack of long-range structure formation spanning the entire dimensions of the specimen. The translucent appearance of the hydrogel is rooted in the presence of these fragments in aqueous solution. The composition of these microstructures and fragments depends on process conditions (reducing agent, solubilizing agent, emulsifier, heat, time, etc.) during isolation from feather. An SEM image of feather keratin hydrogel without allylation (Figure 3b) shows a relatively disordered microstructure whereas the image with allylation (Figure 3c) shows an ordered honeycomb structure, which demonstrates that crosslinking between the random coils of β-keratin chains through the ally groups plays a critical role in structural stability of the β-keratin hydrogel. It further demonstrates that disulfide bridges between the random coils of β-keratin chains play a major role in mechanical properties of the feather [37,38]. 

Human adult mesenchymal stem cells (hMSCs) harvested from the patient’s bone marrow are used extensively in cellular therapies [10,13,44]. Therefore, hMSCs are used to study cell adhesion to the surface of feather β-keratin hydrogels. hMSCs are seeded on the surface of hydrogels and cultured in basal stem cell medium. After seven days, the seeded cells are stained with 4′,6-diamidino-2-phenylindole (DAPI) for imaging cell nuclei, phalloidin for cell cytoskeleton, and vinculin for cell focal adhesions and the result is shown in Figure 3d. According to this figure, there is extensive attachment, spreading and proliferation of hMSCs to the surface of β-keratin hydrogel and the extent of adhesion is similar to that of gelatin (data not shown) [11]. It should be noted that, unlike gelatin, β-keratin chains extracted from feather lack cell-adhesive sequences in their primary structure. The cell adhesion results may be related to the formation of new binding pockets in the β-keratin hydrogel by those microstructures shown in Figure 4. If there was more quantitative information about these binding pockets and the exposed peptides for interaction with receptors on the surface of encapsulated cells, it would potentially lead to wide-spread use of biopolymers isolated from natural tissues in regenerative medicine. Advanced techniques such as total internal reflection microscopy (TIRM) [45], cellular traction force microscopy [46], and fluorescence/bioluminescence resonance energy transfer microscopy (FRET/BRET) [47] should be used in the future to identify and characterize the cell binding pockets in biopolymers isolated from natural tissues.

## 4. Conclusions

Protein-based biopolymers possess a hierarchical structure in their native state which affects their mechanical, as well as biological, properties. These hierarchical structure lead to the formation of long-range structures at the macro-scale spanning the entire dimensions of the natural tissue. Strongly solvating, reducing and stabilizing agents, as well as heat, pressure, and enzymes are used to isolate protein-based biopolymers from their natural tissue, solubilize them in aqueous solution, and convert them into injectable or preformed hydrogels for applications in tissue engineering and regenerative medicine. However, these solubilizing/isolating agents do not completely eliminate the native nano-/micro-structure of the biopolymers. Future work should focus on identifying the nature of secondary, tertiary and high-order structures in hydrogel matrices produced from these biopolymers, quantifying their composition, and characterizing their binding pockets with cell surface receptors. These advances will lead to wide-spread use of protein-based hydrogels derived from natural tissues as injectable or preformed matrices for cell delivery in tissue engineering and regenerative medicine. 

## Figures and Tables

**Figure 1 gels-05-00030-f001:**
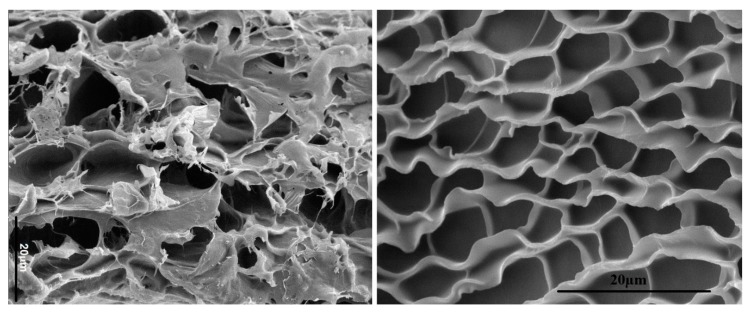
Comparison of the microstructures of decellularized bovine articular cartilage (**left**) and UV-gelled, allylated, digested, decellularized articular cartilage (**right**) as imaged by SEM. Both images show a honeycomb-like microstructure. The scale bars in the images are 20 μm.

**Figure 2 gels-05-00030-f002:**
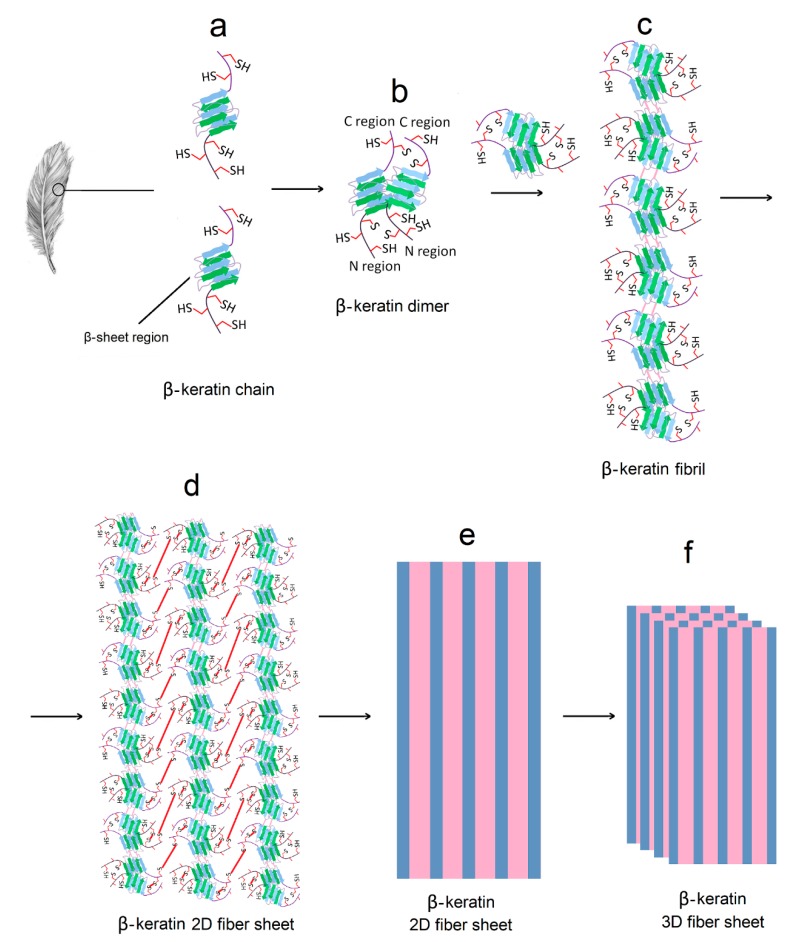
Hierarchical structure of β-keratin in a feather. Extensive secondary interactions including hydrogen bonding, electrostatic and hydrophobic interactions are involved in the formation of β-sheet regions in individual β-keratin molecules (**a**), dimer formation between two β-keratins (**b**), and fibril formation by stacking of β-sheet dimers (**c**). The random coil regions in the 2D fiber sheets (**d** and **e**) and 3D fibrous matrix (**f**) are held together by a network of disulfide (S–S) bridges as well as secondary interactions.

**Figure 3 gels-05-00030-f003:**
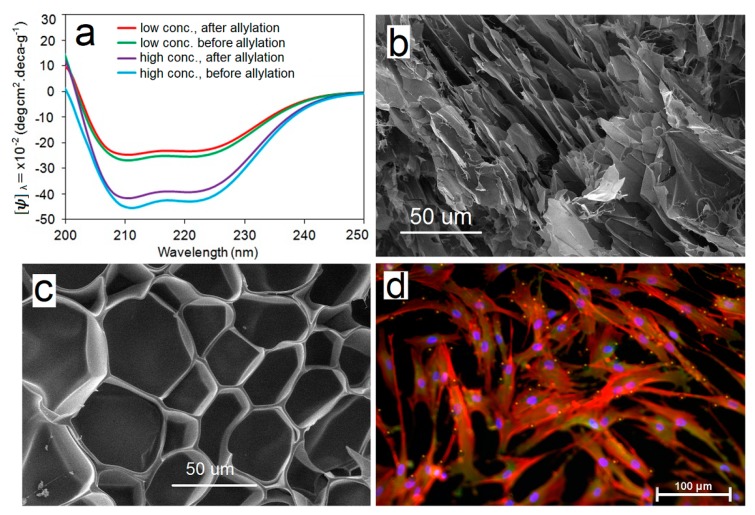
(**a**) Cellular Dichroism (CD) spectra of feather keratin hydrogel at low and high concentrations before and after allylation; SEM image of the freeze-dried feather keratin hydrogel before (**b**) and after (**c**) allylation; (**d**) 4′,6-diamidino-2-phenylindole (DAPI) (purple), phalloidin (red), and vinculin (green) stained images of human adult mesenchymal stem cells (hMSCs) seeded on allylated feather keratin hydrogel after 7 days incubation in basal medium. The bright yellowish dots in (**c**) show focal adhesion points.

**Figure 4 gels-05-00030-f004:**
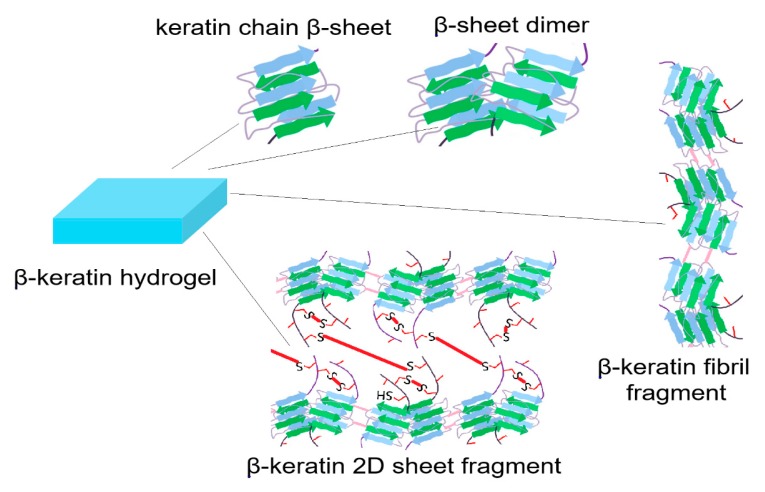
Schematic diagram of the possible microstructures present in the keratin hydrogel derived from feather.

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
