# Peer review of "Challenges for Natural Hydrogels in Tissue Engineering"

_gels, 2019, doi:10.3390/gels5020030_

Round 1

Reviewer 1 Report

In this manuscript, the author used circular dichroism, electron microscopy and cellular studies to elucidate the changes in the microstructure of a protein from native state to a crosslinked hydrogel, and how these changes during processing affect functional of the encapsulated cells. The title of this review is “challenges for nature hydrogels in tissue engineering”, but there are not mentioned in this manuscript. In addition, the reviews should be general and conclusive for existing research in specific field. However, β-keratin is the only one protein mentioned in this review and it does not represent all proteins. Therefore, I do not think this manuscript is suitable to be published as review in the current version.

Below are some comments:

1.      In this manuscript, some methods were chosen to dissolve protein-based materials in aqueous, such as strong solvating, reducing and stabilizing agents, heat and enzymes. What effect does different dissolution approaches have on the structure of the protein?

2.      There are few ways to characterize protein structure, only CD and SEM. Please provide more characterization methods.

3.      For the characterization of protein hydrogels, the manuscript only did cellular study, please provide more characterization of the properties of protein hydrogels.

4.      Only one protein was discussed in this review. Is it consistent with other proteins in structural changes from natural state to hydrogel?

5.      The title is “challenges for nature hydrogels in tissue engineering”, but this manuscript does not mention any existing challenges for nature hydrogels.

Author Response

I have made extensive changes to the manuscript in response to reviewer comments. I would also like to mention that this manuscript is intended as a short review to highlight the need to understand highlight the need to focus on identifying the nature of secondary, tertiary, and higher order structure formation in protein-based hydrogels derived from natural tissues, quantifying their composition, and characterizing their binding pockets with cell surface receptors. This is not intended as a full manuscript but as a review to highlight future direction of research in biopolymers derived from natural tissues. 

Please see the revised manuscript.

Reviewer 2 Report

Gels

Challenges for Natural Hydrogels in Tissue Engineering

Esmaiel Jabbari

This work describes the extraction of β-keratin from bird feather and further processing it with allyl mercaptanol to form a cell encapsulated hydrogel matrix. An attempt is made to study the changes in morphology that the native protein undergoes after being incorporated into a hydrogel matrix. The author also examined the adhesion of hMSCs onto the prepared protein-derived hydrogel despite the hydrogel not possessing any adhesive functional groups.

In Figure 1, it is difficult to see the sulfhydryl groups in a-d because the image is of poor resolution. It is recommended that image of appropriate ppi be included. 

In lines 143-147, the author discusses that the self-assembled structures seen in Figure 2c could be contributing to the honeycomb structure seen in Figure 2b. In addition to not clearly mentioning how the focal adhesion of the hMSCs was visualized (how the optical microscopy images were obtained – i.e., zoom level, lens manufacturer, etc.), the scale bars have been placed rather non-uniformly in 2b (middle bottom center) and 2c (right bottom corner). Additionally, if the author believes that there could be self-assembled structures contributing to the honeycomb design, then it is suggested that a higher magnification of the SEM image must be taken to visualize these structures. The hydrogels can be flash frozen if needed and cross-sections can be obtained for imaging via SEM/FE-SEM.

Disulfide bonds formed by thiol groups are reversible (dissociation energy of 251 kJ/mol). Since the strength of the β-keratin based hydrogel is dependent on these bonds, it is recommended that the author comments on the stability of these hydrogels over a period of time. Specifically, rheological analysis and degradation (tracking weight in physiological pH and salt conditions) could be employed to determine the degradation of the hydrogel under physiological conditions while being subjected to oscillatory strain.  

It would also be beneficial to look at the FTIR spectra of these hydrogels and the actual images of how the β-keratin hydrogel appears in addition to the schematic shown in Figure 3. Since the author has suggested that there are several challenges in the synthesis of naturally derivatized polymers, showing the actual product and some supporting analytical chemistry-based evidence would be useful to other researchers in this field trying to solve similar problems.

Author Response

I have made extensive changes to the manuscript in response to reviewer comments. I like to point out that this review was intended to highlight the need to focus on identifying the nature of secondary, tertiary, and higher order structure formation in protein-based hydrogels derived from natural tissues, quantifying their composition, and characterizing their binding pockets with cell surface receptors. This is not a full research manuscript but a short review to highlight important research that needs to be carried out in biopolymers derived from natural tissues in the future. 

Please see the entire revised manuscript. 

Round 2

Reviewer 1 Report

After extensive revisions by the author, this review is more informative and complete. This review uses model biopolymers, beta-keratin isolated from avian feathers, to illustrate the changes in the microstructure of protein- based biopolymers from their natural state to cross-linked hydrogels, and how these changes affect the function of cells encapsulated in the hydrogel. This will broaden the application of protein hydrogels in tissue engineering and regenerative medicine and provide an effective theoretical basis. Therefore, this review can be accepted in its current form.

Reviewer 2 Report

Significant changes have been made. 

This manuscript is a resubmission of an earlier submission. The following is a list of the peer review reports and author responses from that submission.